# Scalable Multi-view Unsupervised Feature Selection with Structure Learning and Fusion

## ABSTRACT

To tackle the high-dimensional data with multiple representations, multi-view unsupervised feature selection has emerged as a significant learning paradigm. However, previous methods suffer from the following dilemmas: (i) The emphasis is on selecting features to preserve the similarity structure of data, while neglecting the discriminative information in the cluster structure; (ii) They often impose the orthogonal constraint on the pseudo cluster labels, disrupting the locality in the cluster label space; (iii) Learning the similarity or cluster structure from all samples is likewise time-consuming. To this end, a Scalable Multi-view Unsupervised Feature Selection with structure learning and fusion (SMUFS) is proposed to jointly exploit the cluster structure and the similarity relations of data. Specifically, SMUFS introduces the sample-view weights to adaptively fuse the membership matrices that indicate cluster structures and serve as the pseudo cluster labels, such that a unified membership matrix across views can be effectively obtained to guide feature selection. Meanwhile, SMUFS performs graph learning from the membership matrix, preserving the locality of cluster labels and improving their discriminative capability. Further, an acceleration strategy has been developed to make SMUFS scalable for relatively large-scale data. A convergent solution is devised to optimize the formulated problem, and extensive experiments demonstrate the effectiveness and superiority of SMUFS.

## CCS CONCEPTS

• **Computing methodologies → Cluster analysis**; **Dimensionality reduction and manifold learning**.

## KEYWORDS

Multi-view learning; Clustering; Unsupervised feature selection

## 1 INTRODUCTION

As the information technology advances, multi-view data generated from diverse sources or descriptors become common in practical fields, such as multimedia retrieval, image processing and document analysis [15–17, 27, 40]. Despite providing richer feature representations than single-view data, multi-view data are often high-dimensional, in which there inevitably exist low-quality features and even noises [10, 31, 34]. Therefore, the direct use of this kind of data not only involves expensive computation and storage burden

*ACM MM, 2024, Melbourne, Australia*
© 2024 Copyright held by the owner/author(s). Publication rights licensed to ACM.
ACM ISBN 978-x-xxxx-xxxx-x/YY/MM
https://doi.org/10.1145/nnnnnnn.nnnnnnn

but also deteriorates the performance of subsequent tasks. To tackle these problems, multi-view feature selection, aiming to select a compact subset of salient features from heterogeneous feature spaces, becomes a fundamental yet challenging paradigm in data mining. Based on the availability for class labels, existing methods can be performed in supervised, semi-supervised and unsupervised manners [39], respectively. Due to the time-consuming and labor-intensive process for labeling data, unsupervised multi-view feature selection has garnered widespread attention in recent years, which focuses on the intrinsic structures or relations of data and further leverages them to identify discriminative features without the guidance of known labels [19, 21, 43].

To select features from multiple feature spaces, the straightforward way is to concatenate different views first and then perform single-view feature selection on the concatenated feature space. Representative methods include spectral-based feature selection [14, 41, 42] and graph-based feature selection [5, 20, 24], which construct graphs to learn the similarity structures of data and choose features according to the learned structures. Unfortunately, this feature concatenation manner is prone to overlooking the complementarity and correlation among views, such that the effectiveness of selected features might be weakened [6, 13, 36]. Researchers have noticed this issue and introduced weight factors to distinguish different views instead of simply concatenating them. For example, Hou *et al.* [11] utilized view weights to fuse the similarity structures of data in different views, thereby learning a consistent graph for feature selection. Dong *et al.* [7] proposed to learn the collaborative similarity structure by fusing multiple graphs and perform feature selection simultaneously. In [38], Zhang *et al.* exploited both the view-specific projections and consistent projection to learn the similarity structures of data in heterogeneous feature spaces, so as to guide feature selection. Despite leveraging the diversity and correlation among views, these multi-view methods primarily focus on learning graphs to characterize similarity structures and perform feature selection only from the perspective of preserving the similarity information of data. As a result, the cluster structures containing the discriminative information of data are generally overlooked, making selected features hard to distinguish the samples from different cluster centers.

To explore the cluster structure information, Liu *et al.* [18] introduced regression models to learn projection matrices and assumed a linear mapping between projected samples and the cluster indicator matrix (i.e., the pseudo cluster labels), thereby using the pseudo cluster labels provided by $k$-means to supervise the process of unsupervised feature selection. To preserve the consensus and diversity of different views in the cluster label space, Tang *et al.* [25] utilized view-specific projection losses to learn a consensus cluster indicator matrix with the orthogonal and nonnegative constraints for sparse feature selection. To characterize the cluster structure, Fang *et al.* [8] proposed a multi-view unsupervised feature selection method

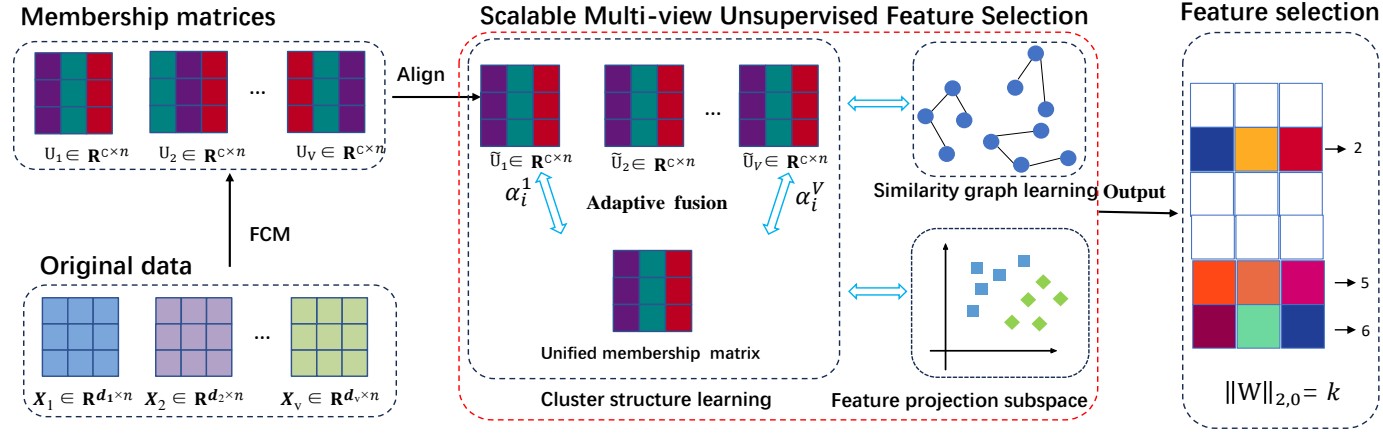

**Figure 1: The illustration of SMUFS. Concretely, SMUFS first employs the membership matrices generated by Fuzzy C-Means (FCM) to explore the view-specific cluster structures. Afterward, the membership matrices are aligned and fused to capture the consistent cluster information across views. Moreover, the similarity relations contained in the membership matrix can be dynamically updated, facilitating feature selection. Finally, the $l_{2,0}$-norm constraint is imposed on the feature projection matrix to identify the top-$k$ features.**

that simultaneously learns orthogonal cluster centers and an shared cluster indicator matrix from projected samples. Generally, these methods first define the feature projection matrices with row sparsity constraints (e.g., the $l_{2,1}$-norm) for feature selection and then perform the pseudo cluster label prediction on projected samples, with the aim to learn the latent cluster information of data. To separate the samples in different clusters, both the orthogonal and nonnegative constraints are imposed on the cluster indicator matrix, which is incapable of guaranteeing nearby data samples having similar cluster labels. To alleviate this issue, Bai *et al.* [1] proposed a multi-view feature selection method that perform the pseudo label learning and graph construction simultaneously, assigning similar samples with similar cluster labels. Shi *et al.* [21] explicitly imposed a binary hash constraint on the cluster label matrix to learn multiple binary labels as the weak supervision information for unsupervised feature selection. Despite making some progress, these methods directly conduct orthogonal clustering on projected samples to explore cluster structure, which is susceptible to low-quality samples due to lacking the guidance of external cluster information, leading to suboptimal performance. Furthermore, to capture the intrinsic structures of data, existing methods directly learn the pseudo cluster labels or similarity graphs from all samples, such that the optimization procedures involves the orthogonal decomposition or inverse operations of high-order matrices, making them unbearable for large-scale data. Another problem is that, the $l_{2,p}$-norm ($0 < p \leq 1$) constraint is often applied to the feature projection matrix for feature selection, introducing an additional parameter to promote the sparsity of feature space. Moreover, the $l_{2,p}$-norm feature selection methods require to calculate the scores of each feature and then rank the feature scores to select features, in the procedures of which slight changes of the feature scores can cause significant fluctuations in feature ranking results [3], impairing the reliability of selected features.

To address the aforementioned issues, a novel method named scalable multi-view unsupervised feature selection with structure learning and fusion (SMUFS) is proposed, whose basic framework

is illustrated in Fig. 1. The main contributions of this paper are summarized as follows:

- We propose a scalable multi-view unsupervised feature selection method, which learns view-specific membership matrices to explore the cluster structure of data and further leverages them to preserve similarity of cluster labels, improving the quality of cluster labels and facilitating feature selection.
- We design an adaptive manner for membership matrices fusion, which integrates the aligned membership matrices from the sample-view perspective to capture the compatible cluster structure across views, not only weakening the negative impacts of poor-quality views and samples but also properly balancing the complementarity and correlation among views.
- We devise an acceleration strategy by learning a bipartite graph between generated anchors and samples, significantly reducing the computational complexity of SMUFS from $O(n^3)$ to $O(nm^2 + m^3)$, so as to efficiently handle large-scale data.

## 2 METHOD

To effectively leverage the underlying structures of the data (i.e., similarity structure and cluster structure), we integrate the adaptive membership fusion, graph learning and feature selection into a unified framework. The details will be elaborated in this section.

## 2.1 Notations and Definitions

Throughout the paper, vectors are denoted in bold lowercase, while matrices are written in bold uppercase. Given an arbitrary matrix $\mathbf{M}$, $\mathbf{m}_i$ and $m_{ij}$ represent its $i$-th row vector and $(i, j)$-th element, and $\text{Tr}(\mathbf{M})$ represents the trace of $\mathbf{M}$. Moreover, $\|\mathbf{M}\|_F = \sqrt{\text{Tr}(\mathbf{M}^T\mathbf{M})}$, $\|\mathbf{M}\|_{2,1} = \sum_i \|\mathbf{m}_i\|_2$ and $\|\mathbf{M}\|_{2,0} = \sum_i \|\mathbf{m}_i\|_2^0$ denote the Frobenius norm, $l_{2,1}$-norm and $l_{2,0}$-norm of $\mathbf{M}$, respectively. $(x)_+ = \max(x, 0)$. Table 1 lists the important notations in this paper.

**Table 1: Description of Notations**

| Notation | Description |
| --- | --- |
| $n$, $m$ | The numbers of samples and anchors, respectively |
| $c$, $V$ | The number of classes and views, respectively |
| $k$ | The number of selected features |
| $d_v$ | The dimension of $v$-th view |
| $d = \sum_{v=1}^{V} d_v$ | The total dimension of $V$ views |
| $\mathbf{x}_i^v \in \mathbb{R}^{d_v \times 1}$ | The $i$-th sample in $v$-th view |
| $\mathbf{x}_i = [\mathbf{x}_i^1, ..., \mathbf{x}_i^V] \in \mathbb{R}^{d \times 1}$ | The $i$-th sample |
| $\mathbf{X} = [\mathbf{x}_1, ..., \mathbf{x}_n] \in \mathbb{R}^{d \times n}$ | The concatenated feature matrix of samples |
| $\{\mathbf{U}_v\}_{v=1}^{V} \in \mathbb{R}^{n \times c}$ | The initialized membership matrices of samples |
| $\{\widetilde{\mathbf{U}}_v\}_{v=1}^{V} \in \mathbb{R}^{n \times c}$ | The aligned membership samples |
| $\mathbf{Z} \in \mathbb{R}^{m \times c}$ | The learned membership matrix of anchors |
| $\mathbf{W} \in \mathbb{R}^{d \times c}$ | The feature selection matrix |
| $\mathbf{1} \in \mathbb{R}^{c \times 1}$ | The all-one vector |

## 2.2 Adaptive Membership Matrix Fusion

The membership matrix, indicating the probability of samples to different clustering centers, can be considered as an effective representation that reflects the data distribution (i.e., soft labels) and contains latent discriminative information [9, 22]. To deal with multi-view data, previous methods typically involve two steps, i.e., generating view-specific membership matrices separately and integrating them directly for further analysis. As the correspondences between cluster centers from multiple views are neglected, different membership matrices are not aligned column-widely [23, 28, 29]. Therefore, the traditional fusion manner without guaranteeing alignment encounter the inconsistent cluster information [35]. Considering that multiple membership matrices should be similar since the relations between samples and cluster centers will not vary with views, the alignment can be achieved by solving the following problem:

$$\min_{\mathbf{P}_v} \|\mathbf{U}_1 - \mathbf{U}_v \mathbf{P}_v\|_F^2 \quad \text{s.t.} \quad \mathbf{P}_v^T \mathbf{P}_v = \mathbf{P}_v \mathbf{P}_v^T = \mathbf{I}, \tag{1}$$

where $\mathbf{P}_v$ represents the permutation matrix of $v$-th view, making $\mathbf{U}_1$ and $\mathbf{U}_v$ consistency in columns (i.e., cluster centers). By decomposing $\mathbf{U}_1^T \mathbf{U}_v$ as $\mathbf{A}\Sigma\mathbf{B}$ (i.e., singular value decomposition), the closed-form solution of Eq. (1) can be calculated as $\mathbf{P}_v = \mathbf{B}\mathbf{A}^T$ [30], and the aligned membership matrix can be written as $\widetilde{\mathbf{U}}_v = \mathbf{U}_v \mathbf{P}_v$ ($v = 1, 2, \cdots, V$).

To learn the distinction and correlation information from data, existing methods usually introduce weight factors to fuse different views, focusing on the information interaction in the view level. However, data collected from heterogeneous sources inevitably contain poor-quality samples, such a fusion manner fails to consider the contribution diversity of samples, impairing the effectiveness of fusion models. To enhance the robustness of multi-view fusion, the differences of samples should be taken into consideration. Thus, an adaptive membership matrix fusion manner devised as follows:

$$\min_{\boldsymbol{\alpha}\mathbf{1}=1, \boldsymbol{\alpha}\geq 0} \sum_{i=1}^{n} \|\mathbf{u}_i - \sum_{v=1}^{V} \alpha_i^v \widetilde{\mathbf{u}}_i^v\|_2^2, \tag{2}$$

where $\mathbf{u}_i$ represents the $i$-th row of the unified membership matrix and $\widetilde{\mathbf{u}}_i^v$ is the $i$-th row of the $v$-th aligned membership matrix. $\boldsymbol{\alpha} = [\boldsymbol{\alpha}_1; \cdots ; \boldsymbol{\alpha}_n] \in \mathbb{R}^{n \times v}$ is the sample-view weight matrix, whose $i$-th row $\boldsymbol{\alpha}_i = [\alpha_i^1, \cdots, \alpha_i^v] \in \mathbb{R}^{1 \times v}$ measures the importance of $i$-sample in each view. In contrast to previous models, Eq. (2) integrates the aligned membership matrices from both the sample and view levels, not only avoiding the effects caused by misalignment but

also discriminating different views and samples, so as to obtain a consistent cluster structure across views.

## 2.3 Feature Selection with Graph Learning

In unsupervised scenarios, learning the pseudo label that reveals the cluster structure and projecting samples to the label space can facilitate the selection of informative features [21]. Therefore, the unified membership matrix that contains the consistent cluster structure information across multiple views can be used as the cluster indicators to guide feature selection, formulated as:

$$\min_{\|\mathbf{W}\|_{2,0}=k} \|\mathbf{X}^T \mathbf{W} - \mathbf{U}\|_F^2. \tag{3}$$

In Eq. (3), the $l_{2,0}$-norm constraint is applied directly rather than the $l_{2,p}$-norm constraint, such that the informative features related to the membership matrix $\mathbf{U}$ can be selected automatically and an additional parameter for the $l_{2,p}$-norm regularization is likewise avoided. To ensure that close samples have similar pseudo labels, we propose graph learning to preserve locality in the learned cluster labels (i.e., the membership matrix):

$$\min_{\mathbf{S}\mathbf{1}=1, \mathbf{S}\geq 0} \sum_{i,j=1}^{n} \|\mathbf{u}_i - \mathbf{u}_j\|_2^2 s_{ij} + \beta \|\mathbf{S}\|_F^2, \tag{4}$$

where $s_{ij}$ measures the similarity between the $i$-th and $j$-th samples, and $\beta$ is the regularization parameter. Eq. (4) assigns larger similarity to the samples that are close to each other in the membership matrix, making the learned membership matrix $\mathbf{U}$ vary smoothly on the graph $\mathbf{S}$. By combining the membership matrix fusion, graph learning and feature selection into a unified framework, the final objective function of SMUFS is derived as follows:

$$\min_{\mathbf{U},\boldsymbol{\alpha},\mathbf{S},\|\mathbf{W}\|_{2,0}=k} \sum_{i=1}^{n} \|\mathbf{u}_i - \sum_{v=1}^{V} \alpha_i^v \widetilde{\mathbf{u}}_i^v\|_2^2 + \lambda \sum_{i,j=1}^{n} \|\mathbf{u}_i - \mathbf{u}_j\|_2^2 s_{ij}$$
$$+ \beta \|\mathbf{S}\|_F^2 + \gamma \|\mathbf{X}^T \mathbf{W} - \mathbf{U}\|_F^2$$
$$\text{s.t.} \quad \boldsymbol{\alpha}\mathbf{1} = 1, \boldsymbol{\alpha} \geq 0, \mathbf{U}\mathbf{1} = 1, \mathbf{U} \geq 0, \mathbf{S}\mathbf{1} = 1, \mathbf{S} \geq 0. \tag{5}$$

In Eq. (5), $\mathbf{U}$ can be dynamically derived from the aligned view-specific membership matrices and the similarity graph $\mathbf{S}$, such that the cluster structure and the similarity structure can promote each other mutually, not only enhancing the discriminative capability of $\mathbf{U}$ but also further facilitating the ultimate feature selection.

## 2.4 Alternate Optimization

Since the $l_{2,0}$-norm constraint is hard to be solved directly, the Augmented Lagrangian Method (ALM) [2] is employed to address it. Specifically, an auxiliary variable $\mathbf{E} = \mathbf{W}$ is first introduced, and thus Eq. (5) can be transformed into the following equivalent form:

$$\min_{\mathbf{U},\boldsymbol{\alpha},\mathbf{S},\mathbf{W},\|\mathbf{E}\|_{2,0}=k} \sum_{i=1}^{n} \|\mathbf{u}_i - \sum_{v=1}^{V} \alpha_i^v \widetilde{\mathbf{u}}_i^v\|_2^2 + \lambda \sum_{i,j=1}^{n} \|\mathbf{u}_i - \mathbf{u}_j\|_2^2 s_{ij}$$
$$+ \beta \|\mathbf{S}\|_F^2 + \gamma \|\mathbf{X}^T \mathbf{W} - \mathbf{U}\|_F^2 + \frac{\mu}{2} \|\mathbf{E} - \mathbf{W} + \frac{\Pi}{\mu}\|_F^2$$
$$\text{s.t.} \quad \boldsymbol{\alpha}\mathbf{1} = 1, \boldsymbol{\alpha} \geq 0, \mathbf{U}\mathbf{1} = 1, \mathbf{U} \geq 0, \mathbf{S}\mathbf{1} = 1, \mathbf{S} \geq 0, \tag{6}$$

where $\mu \in \mathbb{R}^{1 \times 1}$ and $\Pi \in \mathbb{R}^{n \times c}$ represent the penalty parameter and Lagrange multipliers, respectively. Noting that Eq. (6) is not jointly convex with respective to (i.e., w.r.t.) all variables, we design an iterative strategy to archive the optimal solution by alternately optimizing each variable. The optimization procedures are as follows:

• **Update U:** By fixing other variables, the optimization problem of Eq. (6) w.r.t. U becomes:

$$\min_{\mathbf{U}} \sum_{i=1}^{n} \|\mathbf{u}_i - \sum_{v=1}^{V} \alpha_i^v \widetilde{\mathbf{u}}_i^v\|_2^2 + \lambda \sum_{i,j=1}^{n} \|\mathbf{u}_i - \mathbf{u}_j\|_2^2 s_{ij}$$

$$+ \gamma \|\mathbf{X}^T \mathbf{W} - \mathbf{U}\|_F^2 \qquad \text{s.t.} \quad \mathbf{U1} = \mathbf{1}, \mathbf{U} \geq 0. \tag{7}$$

To efficiently solve Eq. (7), a fast two-stage way is adopted. First, we disregard the constrains about U and solve the following problem to obtain the latent solution $\mathbf{U}^*$:

$$\min_{\mathbf{U}^*} \sum_{i=1}^{n} \|\mathbf{u}_i^* - \sum_{v=1}^{V} \alpha_i^v \widetilde{\mathbf{u}}_i^v\|_2^2 + \lambda \sum_{i,j=1}^{n} \|\mathbf{u}_i^* - \mathbf{u}_j^*\|_2^2 s_{ij} + \gamma \|\mathbf{X}^T \mathbf{W} - \mathbf{U}^*\|_F^2. \tag{8}$$

Taking derivative of Eq. (8) w.r.t. $\mathbf{U}^*$ to zero, we have:

$$\mathbf{U}^* = \left( (1+\gamma)\mathbf{I} + \lambda \mathbf{L}_s \right)^{-1} (\mathbf{C} + \gamma \mathbf{X}^T \mathbf{W}), \tag{9}$$

where $\mathbf{L}_s$ is the Laplacian matrix of S, and C denotes the merged membership matrix whose $i$-th row $\mathbf{c}_i = \sum_{v=1}^{V} \alpha_i^v \widetilde{\mathbf{u}}_i^v$. Subsequently, the optimal solution of U can be derived via projecting $\mathbf{U}^*$ into the constrained space, formulated as follows:

$$\min_{\mathbf{U1}=\mathbf{1}, \mathbf{U} \geq 0} \|\mathbf{U} - \mathbf{U}^*\|_F^2, \tag{10}$$

which can be efficiently solved with a closed-form solution [12].

• **Update** $\boldsymbol{\alpha}$: By fixing other variables except $\boldsymbol{\alpha}$, we have the following subproblem:

$$\min_{\boldsymbol{\alpha}\mathbf{1}=\mathbf{1}, \boldsymbol{\alpha} \geq 0} \sum_{i=1}^{n} \|\mathbf{u}_i - \sum_{v=1}^{V} \alpha_i^v \widetilde{\mathbf{u}}_i^v\|_2^2. \tag{11}$$

Considering that Eq. (11) is independent for different rows, so that each row of $\boldsymbol{\alpha}$ (i.e., $\boldsymbol{\alpha}_i$) can be separately optimized. Specifically, the objective function of Eq. (11) can be reformulated as follows (the detailed process is given in Appendix):

$$\min_{\boldsymbol{\alpha}_i \mathbf{1}=\mathbf{1}, \boldsymbol{\alpha}_i \geq 0} \boldsymbol{\alpha}_i \mathbf{D}_i \mathbf{D}_i^T \boldsymbol{\alpha}_i. \tag{12}$$

where $\mathbf{D}_i = [\mathbf{d}_i^1; ...; \mathbf{d}_i^V] \in \mathbb{R}^{V \times c}$, and $\mathbf{d}_i^v = \mathbf{u}_i - \widetilde{\mathbf{u}}_i^v \in \mathbb{R}^{1 \times c}$. Since $\mathbf{D}_i \mathbf{D}_i^T$ is semi-definite, Eq. (12) is a quadratic convex programming problem, which can be solved efficiently.

• **Update W:** When other variables are fixed expect W, the optimization subproblem of Eq. (6) becomes:

$$\min_{\mathbf{W}} \gamma \|\mathbf{X}^T \mathbf{W} - \mathbf{U}\|_F^2 + \frac{\mu}{2} \|\mathbf{E} - \mathbf{W} + \frac{\Pi}{\mu}\|_F^2. \tag{13}$$

Calculating the derivative of Eq. (13) w.r.t. W and setting it to zero, we obtain the optimal solution of W as:

$$\mathbf{W} = (\gamma \mathbf{X} \mathbf{X}^T + \frac{\mu}{2} \mathbf{I})^{-1} (\gamma \mathbf{X} \mathbf{U} + \frac{\mu}{2} \mathbf{E} + \frac{\Pi}{2}). \tag{14}$$

When the number of samples (i.e., $n$) is less than the total feature dimension (i.e., $d$), the matrix identity[1] can be used to reformulate the solution of W, with the aim to reduce the computation complexity:

$$\mathbf{W} = \begin{cases} (\gamma \mathbf{X} \mathbf{X}^T + \frac{\mu}{2} \mathbf{I})^{-1} (\gamma \mathbf{X} \mathbf{U} + \frac{\mu}{2} \mathbf{E} + \frac{\Pi}{2}), & \text{if } d < n \\ (\frac{2}{\mu} \mathbf{I}_d - \frac{4}{\mu^2} \mathbf{X} (\frac{1}{\gamma} \mathbf{I}_n + \frac{2}{\mu} \mathbf{X}^T \mathbf{X})^{-1} \mathbf{X}^T) (\gamma \mathbf{X} \mathbf{U} + \frac{\mu}{2} \mathbf{E} + \frac{\Pi}{2}), & \text{otherwise} \end{cases}. \tag{15}$$

• **Update S:** By fixing other variables except S, we have the following subproblem:

$$\min_{\mathbf{S1}=\mathbf{1}, \mathbf{S} \geq 0} \lambda \|\mathbf{u}_i - \mathbf{u}_j\|_2^2 s_{ij} + \beta \|\mathbf{S}\|_F^2. \tag{16}$$

---
[1] $(\mathbf{A} + \mathbf{C} \mathbf{B} \mathbf{C}^T)^{-1} = \mathbf{A}^{-1} - \mathbf{A}^{-1} \mathbf{C} (\mathbf{B}^{-1} + \mathbf{C}^T \mathbf{A}^{-1} \mathbf{C})^{-1} \mathbf{C}^T \mathbf{A}^{-1}$

Similar to the $\boldsymbol{\alpha}$ subproblem, each row of S (i.e., $\mathbf{s}_i$) is uncorrelated with others, hence Eq. (16) can be optimized for each row independently as follows:

$$\min_{\mathbf{s}_i \mathbf{1}=\mathbf{1}, \mathbf{s}_i \geq 0} \|\mathbf{s}_i + \frac{1}{2\beta} \mathbf{d}_i\|_2^2, \tag{17}$$

where $\mathbf{d}_i$ is a row vector with $d_{ij} = \lambda \|\mathbf{u}_i - \mathbf{u}_j\|_2^2$. Sorting the elements of $\mathbf{d}$ in ascending order to get $\widetilde{\mathbf{d}}$ and assuming that each sample has $f$-nearest neighbors, then the parameter $\beta$ can be determined automatically as: $\beta = \sum_{i=1}^{n} \frac{1}{2n} (k \widetilde{d}_{i,f+1} - \sum_{j=1}^{f} \widetilde{d}_{i,j})$ [20]. The solution of $\mathbf{s}_i$ can be derived as (the detailed process is given in Appendix):

$$s_{ij} = \left( \frac{\widetilde{d}_{i,f+1} - \widetilde{d}_{i,j}}{f \widetilde{d}_{i,f+1} - \sum_{j=1}^{f} \widetilde{d}_{i,j}} \right)_+. \tag{18}$$

• **Update E:** By fixing other variables except E, we have the following subproblem:

$$\min_{\|\mathbf{E}\|_{2,0}=k} \|\mathbf{E} - \mathbf{W} + \frac{\Pi}{\mu}\|_F^2. \tag{19}$$

Denoting $\Theta$ as the set of the $k$ smallest $l_2$-norm row vector of $\mathbf{W} - \frac{\Pi}{\eta}$, then the optimal solution of E can be obtained as:

$$\mathbf{e}_i = \begin{cases} \mathbf{w}_i - \frac{\Pi_i}{\eta}, & \text{if } i \in \Theta; \\ \mathbf{0}, & \text{others} \end{cases} \tag{20}$$

where $\mathbf{e}_i$, $\mathbf{w}_i$ and $\Pi_i$ are the $i$-th rows of E, W and $\Pi$, respectively.

• **Update ALM parameters:** In each iteration, we update the penalty parameter $\mu$ and the Lagrange multipliers $\Pi$ as follows:

$$\Pi = \Pi + \mu (\mathbf{E} - \mathbf{W})$$
$$\mu = \rho \mu. \tag{21}$$

where $\rho$ is a constant update rate. After iteratively solving these subproblems until convergence, we can directly obtain the top-$k$ features from the feature selection matrix W. The entire process for solving Eq. (6) is summarized in Algorithm 1. Now, we further analyze the computation complexity of the proposed SMUFS. Specifically, updating U and W involve the inverse operation of matrices, costing $O(n^3)$ and $O(nd * \min(n,d))$ respectively. When calculating S, it requires $O(n)$ for each row and incurs $O(n^2)$ for the entire S. Besides, the optimization of $\boldsymbol{\alpha}$ and E needs $\text{poly}(V)$ and $O(dk+dc)$. Since $V$ and $c$ are small constants in practice, the main computational complexity of SMUFS is approximated by $O\left(T * (n^3 + n^2 + nd * \min(n,d))\right)$, where $T$ represents the number of iterations.

## 2.5 Accelerated SMUFS with Bipartite Graph

Since SMUFS involves constructing the $n$-order similarity graphs and calculating the inverse of an $n \times n$ dense matrix (i.e., $(1 + \gamma)\mathbf{I} + \lambda \mathbf{L}_s$ in Eq. (10)), it is impractical to handle relatively large-scale problems. Inspired by the bipartite graph strategy [4, 32, 37] that constructs a similarity matrix between representative anchors and samples, an accelerated solution has been devised for SMUFS. Specifically, we firstly employ the FCM to generate $m$ clustering centers as anchors, and then learn a bipartite graph $\mathbf{R} \in \mathbb{R}^{n \times m}$ to explore the similarity relations between samples and anchors. Therefore, the objective function of SMUFS can be reformulated as:

---

**Algorithm 1** : Optimization Algorithm for SMUFS

**Input:** Data $\mathbf{X}$, the cluster number $c$, and the parameters $\lambda$ and $\gamma$;

1: Calculate the view-specific $\mathbf{U}_v$ by FCM and learn the permutation matrices $\mathbf{T}_v$ by solving Eq. (1);
2: Initialize $\alpha_v^i = 1/V$ ($v = 1, \cdots, V$); Initialize $\mathbf{W}$ by $\min \|\mathbf{X}^T\mathbf{W} - \mathbf{U}\|_F^2$; Initialize $\mathbf{S}$ by Eq. (18);
3: **repeat**
4:     Update $\mathbf{U}$ by Eq. (10);
5:     Update $\boldsymbol{\alpha}$ by Eq. (12);
6:     Update $\mathbf{W}$ by Eq. (14);
7:     Update $\mathbf{S}$ by Eq. (18);
8:     Update $\mathbf{E}$ by Eq. (20);
9:     Update $\mu$ and $\Pi$ by Eq. (21);
10: **until** Eq. (6) converges;

**Output:** The projection matrix $\mathbf{W}$ that contains $k$ nonzero rows.

---

$$\min_{\mathbf{U},\mathbf{Z},\boldsymbol{\alpha},\mathbf{R},\mathbf{W},\|\mathbf{E}\|_{2,0}=k} \sum_{i=1}^{n} \|\mathbf{u}_i - \sum_{v=1}^{V} \alpha_i^v \widetilde{\mathbf{u}}_i^v\|_2^2 + \lambda \text{Tr}\left(\begin{bmatrix}\mathbf{U}\\\mathbf{Z}\end{bmatrix}^T \mathbf{L}_{\widetilde{\mathbf{R}}} \begin{bmatrix}\mathbf{U}\\\mathbf{Z}\end{bmatrix}\right)$$

$$+ \beta\|\mathbf{R}\|_F^2 + \gamma\|\mathbf{X}^T\mathbf{W} - \mathbf{U}\|_F^2 + \frac{\mu}{2}\|\mathbf{E} - \mathbf{W} + \frac{\Pi}{\mu}\|_F^2$$

$$\text{s.t. } \boldsymbol{\alpha}\mathbf{1} = 1, \boldsymbol{\alpha} \geq 0, \mathbf{U}\mathbf{1} = 1, \mathbf{U} \geq 0, \mathbf{Z}\mathbf{1} = 1, \mathbf{Z} \geq 0, \mathbf{R}\mathbf{1} = 1, \mathbf{R} \geq 0. \quad (22)$$

where $\mathbf{Z} \in \mathbb{R}^{m \times c}$ represents the learned membership matrix of anchors, and $\mathbf{L}_{\widetilde{\mathbf{R}}} \in \mathbb{R}^{(n+m) \times (n+m)}$ denotes the Laplacian matrix of the augmented graph $\widetilde{\mathbf{R}} = \begin{bmatrix}\mathbf{0} & \mathbf{R}\\\mathbf{R}^T & \mathbf{0}\end{bmatrix}$, calculated as:

$$\mathbf{L}_{\widetilde{\mathbf{R}}} = \begin{bmatrix}\mathbf{D} & \mathbf{0}\\\mathbf{0} & \boldsymbol{\Lambda}\end{bmatrix} - \begin{bmatrix}\mathbf{0} & \mathbf{R}\\\mathbf{R}^T & \mathbf{0}\end{bmatrix} = \begin{bmatrix}\mathbf{I}_n & -\mathbf{R}\\-\mathbf{R}^T & \boldsymbol{\Lambda}\end{bmatrix}, \quad (23)$$

where $\mathbf{D}$ and $\boldsymbol{\Lambda}$ are diagonal matrices whose elements are row sums and column sums of the bipartite graph $\mathbf{R}$, respectively. Noting that the subproblems of $\mathbf{W}$, $\boldsymbol{\alpha}$ and $\mathbf{E}$ remain consistent with the Eq. (6), indicating that they can be directly solved using the corresponding steps in Algorithm 1. When updating $\mathbf{R}$, the subproblem is:

$$\min_{\mathbf{R}\mathbf{1}=1,\mathbf{R}\geq 0} \sum_{i=1}^{n} \sum_{j=1}^{m} \|\mathbf{u}_i - \mathbf{z}_j\|_2^2 r_{ij} + \beta\|\mathbf{R}\|_F^2, \quad (24)$$

which can be computed by rows, requiring the computational complexity of $O(nm)$. To optimize $\mathbf{U}$ and $\mathbf{Z}$, the subproblem of Eq. (22) is transformed into:

$$\min_{\mathbf{U},\mathbf{Z}} \|\mathbf{U} - \mathbf{C}\|_F^2 + \lambda \text{Tr}\left(\begin{bmatrix}\mathbf{U}\\\mathbf{Z}\end{bmatrix}^T \mathbf{L}_{\widetilde{\mathbf{R}}} \begin{bmatrix}\mathbf{U}\\\mathbf{Z}\end{bmatrix}\right) + \gamma\|\mathbf{X}^T\mathbf{W} - \mathbf{U}\|_F^2$$

$$\text{s.t. } \mathbf{U}\mathbf{1} = 1, \mathbf{U} \geq 0, \mathbf{Z}\mathbf{1} = 1, \mathbf{Z} \geq 0. \quad (25)$$

Setting the derivative of Eq. (25) w.r.t. $\mathbf{Z}$ to zero directly, the latent solution of $\mathbf{Z}$ is derived as:

$$\mathbf{Z}^* = \boldsymbol{\Lambda}^{-1}\mathbf{R}^T\mathbf{U}. \quad (26)$$

Noting that $\mathbf{Z}^*$ satisfies the constrains of $\mathbf{Z}$ (i.e., $\boldsymbol{\Lambda}^{-1}\mathbf{R}^T\mathbf{U}\mathbf{1} = 1$ and $\boldsymbol{\Lambda}^{-1}\mathbf{R}^T\mathbf{U} > 0$), thus $\mathbf{Z}^*$ is the optimal solution of Eq. (25). Then, substituting $\mathbf{Z} = \boldsymbol{\Lambda}^{-1}\mathbf{R}^T\mathbf{U}$ into Eq. (25) and setting its derivative w.r.t. $\mathbf{U}$ to zero, the latent solution of $\mathbf{U}^*$ is obtained as:

$$\mathbf{U}^* = (\mathbf{H} - \lambda\mathbf{R}\boldsymbol{\Lambda}^{-1}\mathbf{R}^T)^{-1}(\mathbf{C} + \gamma\mathbf{X}^T\mathbf{W}), \quad (27)$$

where $\mathbf{H} = (1 + \lambda + \gamma)\mathbf{I}$. Noting that Eq. (27) also involves the inverse operation of an $n \times n$ dense matrix (i.e., $\mathbf{H} - \lambda\mathbf{R}\boldsymbol{\Lambda}^{-1}\mathbf{R}^T$), requiring the

**Table 2: Detailed information on multi-view datasets.**

| Dataset | Classes | Data size | Feature size |
|---------|---------|-----------|--------------|
| MSRC-v1 | 7 | 210 | 2418(1302/48/512/100/256/200) |
| ORL | 40 | 400 | 1689(512/59/864/254) |
| Cal-9 | 9 | 900 | 3766(48/40/254/1984/512/928) |
| COIL-20 | 20 | 1440 | 2801(512/420/1239/630) |
| Leaves | 100 | 1600 | 192(64/64/64) |
| HW | 10 | 2000 | 649(240/76/216/47/64/6) |
| Voxceleb | 50 | 18354 | 757(179/475/21/17/65) |
| Mnist | 10 | 20000 | 459(256/144/59) |

computation complexity of $O(n^3)$ at least. To this end, we further simplifies the solution of $\mathbf{U}^*$ by the matrix identity[2]:

$$\mathbf{U}^* = \mathbf{H}^{-1}(\mathbf{C} + \gamma\mathbf{X}^T\mathbf{W}) + \mathbf{H}^{-1}\mathbf{R}(\frac{\boldsymbol{\Lambda}}{\lambda} - \mathbf{R}^T\mathbf{H}^{-1}\mathbf{R})^{-1}\mathbf{R}^T\mathbf{H}^{-1}(\mathbf{C} + \lambda\mathbf{X}^T\mathbf{W}). \quad (28)$$

Therefore, the inverse of $\mathbf{H} - \lambda\mathbf{R}\boldsymbol{\Lambda}^{-1}\mathbf{R}^T$ is equivalently substituted by the inverses of a diagonal matrix (i.e., $\mathbf{H}$) and an $m \times m$ matrix (i.e., $\frac{\boldsymbol{\Lambda}}{\lambda} - \mathbf{R}^T\mathbf{H}^{-1}\mathbf{R}$). By calculating the above formula from right to left, the computation complexity of solving $\mathbf{U}^*$ can be reduced from $O(n^3)$ to $O(nm^2 + m^3)$. Considering that $m \ll n$ for large-scale data, our accelerated strategy can reduce the main computation complexity of SMUFS from $O(n^3 + n^2 + nd * \min(n, d))$ to $O(nm^2 + nd * \min(n, d) + m^3)$.

## 3 EXPERIMENTS

### 3.1 Experimental Settings

In this section, eight real-word datasets are employed, including: MSRC-v1[3], ORL[4], Cal-9[5], Leaves[6], COIL-20[7], HW[8], Mnist[9] and Voxceleb[10]. The details of datasets are summarized in Table 2. To fully evaluate the effectiveness and efficiency of SMUFS, we conduct the comparative experiments with five state-of-the-art feature selection methods, including (1) Multi-view Unsupervised Feature Selection with Adaptive Similarity and View Weight (**ASVW**) [11]; (2) Multilevel Projections with Adaptive Neighbor Graph for Unsupervised Multi-View Feature Selection (**MAMFS**) [38]; (3) Robust Unsupervised Feature Selection via Multi-Group Adaptive Graph Representation (**MGAGR**) [33]. (4) Unsupervised Feature Selection with Binary Hashing (**FSBH**) [21]. (5) Joint Multi-View Unsupervised Feature Selection and Graph Learning (**JMVFS**) [8]. To ensure fair comparisons, the parameters of all compared methods are searched according to their respective works. For the proposed SMUFS, the regularization parameters $\lambda$ and $\gamma$ are tuned from $\{10^{-3}, 10^{-2}, \cdots, 10^3\}$. To enhance the efficiency of SMUFS on the Mnist and Voxceleb datasets, the acceleration strategy proposed in Section 2.5 is employed to construct bipartite graphs by setting the number of anchors as $m = 10\% \times n$. After obtaining feature subsets by each feature selection method, K-means clustering is executed 20 times independently on the samples represented by the selected

---

[2] $(\mathbf{A} + \mathbf{CBC}^T)^{-1} = \mathbf{A}^{-1} - \mathbf{A}^{-1}\mathbf{C}(\mathbf{B}^{-1} + \mathbf{C}^T\mathbf{A}^{-1}\mathbf{C})^{-1}\mathbf{C}^T\mathbf{A}^{-1}$
[3] http://research.microsoft.com/en-us/projects/objectclassrecognition/
[4] https://www.kaggle.com/datasets/tavarez/the-orl-database-for-training-and-testing
[5] https://data.caltech.edu/records/mzrjq-6wc02
[6] https://archive.ics.uci.edu/dataset/
[7] https://www.cs.columbia.edu/CAVE/software/softlib/coil-20.php
[8] https://archive.ics.uci.edu/dataset/72/multiple+features
[9] https://www.kaggle.com/datasets/hojjatk/mnist-dataset
[10] https://www.robots.ox.ac.uk/~vgg/data/voxceleb/

**Table 3: ACC of different methods with different numbers of features. The best and second results are in bold and underlined.**

| Datasets | Feature ratio | 10% | 15% | 20% | 25% | 30% | 35% |
|---|---|---|---|---|---|---|---|
| MSRC-v1 | ASVW | 78.67±1.74 | 80.81±0.83 | 81.00±0.62 | 82.10±1.26 | 82.43±1.51 | 81.95±1.48 |
| | MAMFS | 66.76±3.96 | 65.62±3.94 | 65.67±3.10 | 65.95±4.43 | 66.81±5.37 | 69.71±6.42 |
| | MGAGR | 84.71±6.05 | 89.29±4.62 | 90.57±3.63 | 90.71±4.06 | 92.19±1.45 | 92.19±3.97 |
| | FSBH | 91.95±2.79 | 92.48±1.50 | 92.86±0.98 | 92.38±1.00 | 91.86±0.75 | 92.95±1.08 |
| | JMVFS | 94.24±0.69 | 94.00±0.56 | 94.67±1.17 | 94.43±1.03 | 94.10±0.56 | 93.67±0.61 |
| | SMUFS | **97.43**±0.49 | **96.19**±1.02 | **95.71**±0.82 | **96.05**±0.85 | **95.62**±0.92 | **94.76**±0.48 |
| ORL | ASVW | 65.05±1.71 | 64.90±1.68 | 68.28±2.34 | 67.98±2.96 | 69.90±3.90 | 70.53±2.45 |
| | MAMFS | 64.17±2.47 | 63.25±2.14 | 63.85±3.27 | 64.20±2.47 | 64.18±2.04 | 64.15±3.89 |
| | MGAGR | 65.35±4.07 | 67.10±2.38 | 67.55±2.10 | 68.90±2.44 | 67.90±3.19 | 68.58±2.96 |
| | FSBH | 67.30±2.09 | 64.97±2.28 | 65.85±2.35 | 65.33±3.23 | 65.78±2.59 | 65.33±3.37 |
| | JMVFS | 74.58±3.01 | **74.32**±3.42 | 72.63±2.39 | 73.38±3.35 | 74.00±2.21 | 72.78±2.35 |
| | SMUFS | **74.93**±2.71 | 74.02±2.08 | **72.73**±2.67 | **75.87**±1.62 | **74.08**±2.43 | **74.48**±2.87 |
| Cal-9 | ASVW | 72.40±0.74 | 74.01±1.29 | 74.63±1.06 | 74.50±1.12 | 74.42±0.54 | 74.23±0.52 |
| | MAMFS | 69.71±0.65 | 69.98±1.13 | 71.26±1.16 | 72.12±0.37 | 72.99±0.77 | 73.12±0.61 |
| | MGAGR | 72.60±1.63 | 73.64±1.25 | 74.57±0.68 | 75.29±1.01 | 75.01±0.48 | 74.89±1.06 |
| | FSBH | 72.86±1.47 | 74.68±1.71 | 75.69±0.76 | 75.48±1.03 | 75.08±0.61 | 75.46±0.44 |
| | JMVFS | 75.09±0.61 | 75.49±0.55 | 76.03±0.58 | 75.64±0.39 | **76.21**±0.47 | 76.16±0.64 |
| | SMUFS | **76.08**±1.03 | **76.04**±0.59 | **76.22**±0.58 | **76.01**±0.76 | 76.12±0.46 | **76.22**±0.79 |
| Leaves | ASVW | 46.16±1.04 | 53.62±1.33 | 58.54±1.06 | 62.44±1.67 | 66.39±2.12 | 68.49±1.55 |
| | MAMFS | 50.19±0.99 | 54.98±1.13 | 59.79±2.21 | 63.64±1.46 | 65.93±1.95 | 67.31±1.34 |
| | MGAGR | 43.71±0.52 | 54.20±1.76 | 59.78±1.18 | 64.93±1.77 | 67.93±1.83 | 70.79±1.61 |
| | FSBH | 56.54±1.06 | **61.53**±1.44 | 65.79±1.66 | 67.75±0.81 | 67.19±1.38 | 68.58±1.17 |
| | JMVFS | 52.95±0.89 | 61.26±1.29 | 66.27±1.70 | 68.98±2.02 | 70.20±1.16 | 73.56±1.81 |
| | SMUFS | **57.01**±1.44 | 61.29±1.36 | **66.99**±1.56 | **69.76**±1.74 | **70.92**±1.73 | **74.91**±1.53 |
| COIL20 | ASVW | 73.22±2.34 | 74.51±2.29 | 74.76±2.55 | 73.68±3.23 | 75.13±3.15 | **77.25**±2.53 |
| | MAMFS | 75.77±2.12 | 76.71±2.76 | 75.20±2.32 | 75.49±3.51 | 75.69±2.66 | 75.76±1.55 |
| | MGAGR | 66.22±2.15 | 69.77±2.57 | 70.89±2.44 | 71.17±3.91 | 72.15±5.19 | 71.29±3.18 |
| | FSBH | 74.45±2.62 | 74.40±2.89 | 75.06±3.85 | 74.50±2.46 | 76.29±4.42 | 74.01±3.67 |
| | JMVFS | 76.49±2.74 | 76.83±3.83 | 75.53±2.55 | 76.55±2.29 | 76.42±3.86 | 76.01±2.69 |
| | SMUFS | **77.02**±1.99 | **76.97**±2.18 | **75.78**±3.80 | **76.70**±3.28 | **76.75**±2.90 | 76.29±3.00 |
| HW | ASVW | 92.23±1.05 | 93.28±0.18 | 93.09±0.10 | 94.91±1.78 | 93.13±0.14 | 94.76±3.18 |
| | MAMFS | 82.08±3.72 | 89.09±0.11 | 89.75±0.47 | 89.87±0.23 | 89.96±0.24 | 90.75±0.16 |
| | MGAGR | 65.06±2.24 | 69.44±3.97 | 82.56±0.81 | 89.90±0.10 | 91.71±0.22 | 91.05±0.13 |
| | FSBH | 87.68±0.56 | 88.75±0.39 | 90.29±0.24 | 90.87±0.26 | 91.01±0.29 | 92.04±0.13 |
| | JMVFS | **92.35**±0.12 | 94.21±0.08 | 93.82±0.10 | 94.11±0.08 | 94.08±0.17 | 93.88±1.34 |
| | SMUFS | 91.85±1.25 | **94.38**±0.15 | **94.07**±0.56 | **95.06**±0.28 | **94.30**±0.31 | **94.81**±0.21 |
| Voxceleb | ASVW | 75.66±2.93 | 76.00±2.88 | 76.23±2.69 | 77.90±2.59 | 77.83±4.06 | **80.45**±2.15 |
| | MAMFS | 58.76±1.69 | 72.52±3.45 | 76.12±2.51 | 79.85±2.35 | 80.08±3.44 | 80.11±3.23 |
| | MGAGR | 75.26±2.65 | 76.67±3.20 | 77.26±1.01 | 78.45±2.01 | 78.57±1.67 | 79.06±1.83 |
| | FSBH | **77.53**±2.08 | 77.88±3.52 | 78.86±3.94 | 79.17±2.34 | 80.45±3.76 | 80.14±2.96 |
| | JMVFS | 76.37±2.28 | **78.72**±1.48 | **79.08**±2.81 | 78.85±1.34 | 80.17±3.35 | 79.37±1.78 |
| | SMUFS | 77.21±1.83 | 78.55±2.46 | 78.94±2.25 | **80.11**±3.29 | **80.51**±1.31 | **80.57**±2.61 |
| Mnist | ASVW | 30.16±0.17 | 32.20±0.91 | 34.40±0.57 | 30.85±0.46 | 31.26±0.58 | 32.49±0.30 |
| | MAMFS | 32.46±0.47 | 34.33±0.55 | 33.74±1.85 | 32.62±1.70 | 33.30±2.34 | 32.06±0.60 |
| | MGAGR | 23.97±0.71 | 25.66±0.38 | 35.17±2.34 | **37.43**±2.53 | **34.04**±3.02 | 37.29±2.80 |
| | FSBH | **35.65**±0.48 | 37.80±1.85 | 35.15±1.63 | 35.68±0.12 | 33.43±2.08 | 33.21±1.69 |
| | JMVFS | 34.68±0.09 | 36.76±0.71 | 35.52±1.92 | 36.44±1.11 | 33.94±0.21 | 36.98±0.25 |
| | SMUFS | 35.36±0.46 | **38.30**±0.69 | **35.66**±1.55 | 36.23±1.48 | 33.58±1.98 | **37.82**±1.64 |

features. The average results, including accuracy (ACC) and normalized mutual information (NMI), are reported to evaluate the quality of the selected features by different methods.

## 3.2 Comparison Results

Tables 3 and 4 present the means and standard deviations of ACC and NMI, where the optimal and sub-optimal results are highlighted in bold and underlined, respectively. From the results, we can drawn the following conclusions: (1) As the number of selected features varies, SMUFS consistently achieves competitive or superior results to others, indicating its effectiveness when handle multi-view unsupervised feature selection tasks. (2) The ACC and NMI obtained by SMUFS outperform the methods that focus on selecting features to preserve the similarity structure (i.e., ASVW, MAMFS and MGAGR), underscoring the importance of leveraging discriminative information in the cluster structure. (3) Compared to the methods

that consider both cluster structure and similarity structure (i.e., FSBH and JMVFS), SMUFS exhibits better performance in most cases, emphasizing the significance of integrating external cluster information and preserving the locality of cluster labels. Meanwhile, Table 5 presents the training time of each method on small datasets, validating that SMUFS can obtain comparable efficiency to other methods. And Fig. 2 further depicts the relation between the running time and the data size on the Voxceleb and Mnist datasets. As the size of the training samples increase, we observe that the training time of other methods rises exponentially, whereas the accelerated SUMFS exhibits a linear growth, demonstrating the scalability of SMUFS when dealing with large datasets.

**Table 4: NMI of different methods with different numbers of features. The best and second results are in bold and underlined.**

| Datasets | Feature ratio | 10% | 15% | 20% | 25% | 30% | 35% |
|---|---|---|---|---|---|---|---|
| MSRC-v1 | ASVW | 68.37±2.30 | 71.36±0.76 | 71.14±0.41 | 70.97±1.37 | 71.74±1.81 | 71.88±1.44 |
| | MAMFS | 63.09±1.48 | 62.39±1.28 | 62.23±1.68 | 63.05±1.23 | 63.15±1.76 | 64.32±2.07 |
| | MGAGR | 78.67±5.61 | 84.40±3.36 | 85.53±2.82 | 85.43±3.68 | 86.54±2.10 | 87.05±3.56 |
| | FSBH | 85.19±3.66 | 85.78±3.21 | 86.37±3.09 | 86.40±2.78 | 86.02±4.72 | 87.60±1.66 |
| | JMVFS | 87.75±3.26 | 87.51±0.82 | 86.77±1.09 | 85.59±1.34 | 86.41±1.12 | 86.11±0.83 |
| | SMUFS | 94.68±0.91 | 92.25±1.59 | 91.32±1.36 | 92.15±1.33 | 91.12±1.60 | 89.70±2.18 |
| ORL | ASVW | 82.71±1.14 | 83.12±0.62 | 84.86±1.27 | 85.73±1.54 | 86.75±1.39 | 87.50±1.16 |
| | MAMFS | 81.97±1.18 | 81.02±0.79 | 81.51±1.25 | 82.05±1.44 | 81.73±1.12 | 82.47±1.44 |
| | MGAGR | 83.74±1.69 | 84.22±1.22 | 85.09±1.42 | 86.21±1.53 | 85.29±1.17 | 85.54±1.30 |
| | FSBH | 83.66±1.72 | 83.16±1.94 | 83.17±1.33 | 83.31±1.45 | 83.09±1.48 | 83.35±0.99 |
| | JMVFS | 89.93±0.87 | 90.35±1.26 | 89.28±1.13 | 89.55±1.09 | 89.11±1.23 | 88.78±1.07 |
| | SMUFS | 90.22±1.13 | 89.57±1.16 | 89.81±1.18 | 90.90±1.07 | 89.81±0.99 | 89.94±0.98 |
| Cal-9 | ASVW | 59.32±0.46 | 61.70±0.36 | 63.22±0.50 | 64.17±0.34 | 64.54±0.49 | 64.52±0.45 |
| | MAMFS | 59.67±0.65 | 60.38±0.62 | 60.87±0.54 | 61.68±1.09 | 62.30±0.61 | 62.71±0.55 |
| | MGAGR | 62.84±0.98 | 63.68±1.19 | 64.59±0.75 | 65.60±1.10 | 65.75±0.49 | 65.08±0.77 |
| | FSBH | 61.55±1.13 | 65.66±0.70 | 66.46±0.77 | 66.96±0.89 | 66.71±0.48 | 67.37±0.53 |
| | JMVFS | 64.67±0.31 | 65.45±0.32 | 66.06±0.41 | 65.67±0.58 | 66.39±0.65 | 66.42±0.58 |
| | SMUFS | 66.34±0.65 | 66.36±0.52 | 66.18±0.72 | 67.05±0.79 | 66.83±0.59 | 66.48±0.83 |
| Leaves | ASVW | 71.41±0.32 | 76.90±0.40 | 79.98±0.41 | 82.72±0.77 | 85.14±0.62 | 87.20±0.53 |
| | MAMFS | 74.58±0.51 | 77.26±0.55 | 79.77±0.71 | 82.46±0.45 | 83.85±0.53 | 84.94±0.44 |
| | MGAGR | 69.45±0.43 | 76.41±0.63 | 80.57±0.40 | 83.92±0.70 | 85.72±0.47 | 87.51±0.71 |
| | FSBH | 78.14±0.44 | 82.69±0.65 | 86.03±0.49 | 87.26±0.43 | 88.49±0.40 | 89.00±0.31 |
| | JMVFS | 75.70±0.29 | 81.65±0.55 | 85.75±0.35 | 86.63±0.71 | 88.30±0.42 | 89.26±0.54 |
| | SMUFS | 78.96±0.46 | 81.35±0.53 | 86.10±0.52 | 86.91±0.48 | 88.76±0.57 | 90.11±0.76 |
| COIL20 | ASVW | 83.81±1.25 | 84.28±1.02 | 85.35±1.20 | 84.73±1.23 | 85.21±1.40 | 86.74±1.63 |
| | MAMFS | 84.48±1.40 | 85.24±0.93 | 84.78±1.60 | 85.37±1.30 | 85.39±1.51 | 85.43±0.79 |
| | MGAGR | 81.02±2.10 | 82.78±0.94 | 83.22±1.16 | 84.31±1.91 | 84.63±1.76 | 84.53±2.07 |
| | FSBH | 84.86±0.58 | 85.22±1.60 | 85.69±1.20 | 85.90±1.22 | 87.43±1.58 | 85.48±2.22 |
| | JMVFS | 86.67±1.44 | 85.80±1.13 | 86.21±1.28 | 86.31±1.07 | 87.55±1.12 | 86.37±1.61 |
| | SMUFS | 87.57±1.48 | 86.89±0.74 | 86.56±1.14 | 87.24±1.20 | 86.70±1.07 | 86.61±1.50 |
| HW | ASVW | 85.30±0.91 | 89.05±0.71 | 88.46±1.47 | 89.60±0.29 | 87.93±1.67 | 89.40±0.14 |
| | MAMFS | 77.87±1.79 | 81.29±0.20 | 82.31±0.66 | 83.58±0.21 | 83.87±1.87 | 84.35±0.30 |
| | MGAGR | 48.29±1.40 | 54.88±2.02 | 70.46±0.70 | 80.61±0.18 | 83.82±0.23 | 83.77±0.26 |
| | FSBH | 78.21±0.19 | 80.41±0.29 | 82.26±0.34 | 83.63±2.77 | 83.51±0.32 | 84.85±1.19 |
| | JMVFS | 89.24±0.20 | 88.54±0.09 | 88.04±0.14 | 88.96±0.16 | 88.93±2.03 | 89.18±1.88 |
| | SMUFS | 88.18±0.80 | 89.65±0.16 | 90.08±1.36 | 91.14±0.25 | 89.36±1.54 | 89.67±0.24 |
| Voxceleb | ASVW | 86.28±1.00 | 88.72±1.12 | 89.39±0.84 | 90.39±1.08 | 90.77±1.30 | 91.91±0.62 |
| | MAMFS | 71.79±0.81 | 86.87±1.02 | 89.19±0.92 | 90.41±0.68 | 91.41±0.39 | 92.18±1.07 |
| | MGAGR | 82.33±0.87 | 84.11±0.58 | 85.83±0.74 | 86.99±0.75 | 89.98±1.22 | 88.12±1.04 |
| | FSBH | 86.84±0.82 | 87.64±1.09 | 87.74±1.52 | 88.48±0.85 | 89.20±0.95 | 89.39±0.96 |
| | JMVFS | 87.44±0.79 | 89.72±0.58 | 90.57±0.88 | 91.32±1.08 | 91.77±0.78 | 91.52±0.99 |
| | SMUFS | 86.94±0.35 | 89.65±0.72 | 90.95±0.74 | 91.46±0.57 | 91.52±0.54 | 91.81±0.94 |
| Mnist | ASVW | 19.10±0.11 | 21.18±0.35 | 22.42±0.59 | 20.48±0.21 | 21.51±0.62 | 22.87±0.38 |
| | MAMFS | 22.78±0.44 | 25.38±0.90 | 25.29±1.04 | 25.05±0.60 | 25.32±1.65 | 24.33±0.29 |
| | MGAGR | 12.18±0.29 | 17.04±0.37 | 26.28±1.36 | 27.54±1.44 | 26.73±2.08 | 28.09±1.92 |
| | FSBH | 22.20±0.74 | 25.39±1.33 | 21.94±1.31 | 23.89±0.24 | 23.50±0.97 | 23.82±1.31 |
| | JMVFS | 23.59±0.24 | 24.22±0.36 | 25.97±0.98 | 24.06±0.31 | 26.14±2.05 | 23.26±0.03 |
| | SMUFS | 25.88±0.65 | 26.02±1.08 | 28.07±1.27 | 26.61±1.32 | 25.93±1.07 | 27.51±1.35 |

**Table 5: Running time (in seconds) of each method. The best and second results are in bold and underlined, respectively.**

| Datasets | ASVW | MAMFS | MGAGR | FSBH | JMVFS | SMUFS |
|---|---|---|---|---|---|---|
| MSRC-v1 | 1.83 | 9.89 | 1.48 | 2.48 | **0.16** | 0.45 |
| ORL | 1.53 | 4.05 | 1.61 | 1.42 | **0.23** | 0.27 |
| Cal-9 | 4.60 | 37.36 | 20.08 | 9.93 | 1.42 | **1.21** |
| Leaves | 0.63 | 0.52 | 2.13 | **0.31** | 1.44 | 0.40 |
| COIL20 | 2.41 | 14.11 | 31.17 | 6.53 | 1.55 | **0.93** |
| HW | 0.91 | 2.04 | 9.41 | 7.68 | 3.27 | **0.55** |

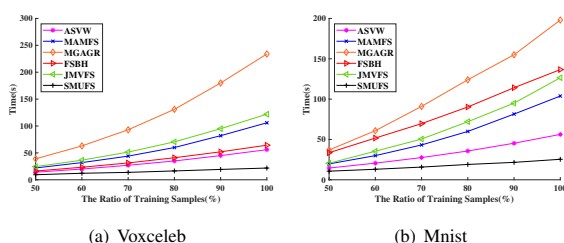

(a) Voxceleb      (b) Mnist

**Figure 2: Running time versus the number of samples.**

## 3.3 Ablation Study

To verify the effectiveness of each components in the proposed structure learning and fusion model, an ablation experiment is conducted. Specifically, three variants of SMUFS have been designed: $SMUFS_1$ neglects the correspondence of clustering centers from multiple views, learning the cluster information from unaligned membership matrices; $SMUFS_2$ directly employs the view weights to discriminate multiple views and fuse membership matrices only from the view perspective; $SMUFS_3$ retains membership matrix fusion and excludes the graph learning part. The experimental results of SMUFS and its simplified versions are presented in Fig. 3, where

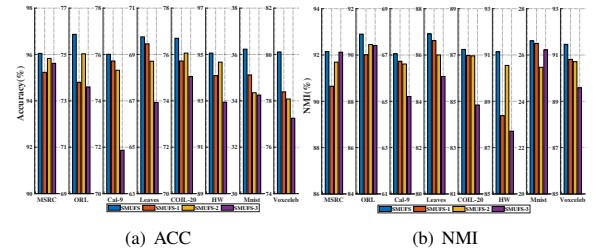

(a) ACC

(b) NMI

**Figure 3: ACC and NMI of SMUFS and its simplified versions.**

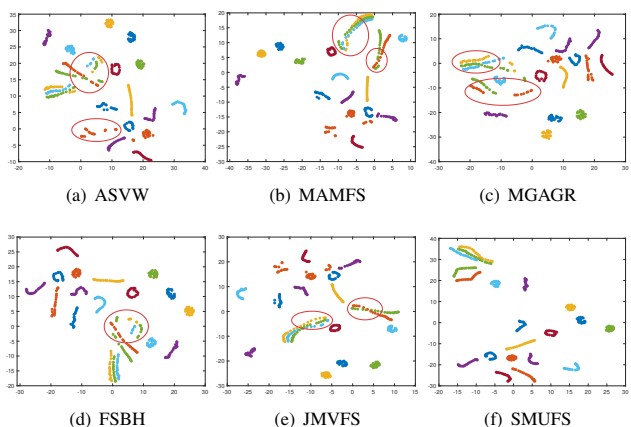

(a) ASVW

(b) MAMFS

(c) MGAGR

(d) FSBH

(e) JMVFS

(f) SMUFS

**Figure 4: The T-SNE visualizations on the COIL-20 dataset using different feature subsets selected by each method.**

the number of selected features is fixed as $25\% \times d$. We can conclude that: (1) Comparative results between SMUFS and SMUFS$_1$ validate that directly fusing membership matrices without alignment compromises the consistency of cluster structures among views. (2) SMUFS consistently outperforms SMUFS$_2$, demonstrating that discriminating different samples can reduce the impact of outliers and facilitate the ultimate feature selection. (3) The clustering performance of SMUFS is superior to SMUFS$_3$, indicating that dynamic graph learning preserves the locality of indicator labels and exerts a crucial influence on the overall performance.

## 3.4 Visualization

To intuitively demonstrate the quality of selected features, we apply the T-SNE method [26] to project the high-dimensional feature space in a two-dimensional space. Specifically, 400 samples from 20 clusters of COIL20 dataset are selected for visualization, and Fig. 4 illustrates the results of feature subsets selected by each method, in which the number of selected features is fixed at $25\% \times d$. As displayed in Fig. 4, we can observe that the feature subset selected by SMUFS can effectively separate samples from different clustering centers, while the visualizations of others exhibit different degrees of overlaps. Furthermore, the distances between clustering centers obtained by SMUFS are larger compared to others, indicating that utilizing the external cluster information can benefit the feature selection process as well as facilitate subsequent clustering tasks.

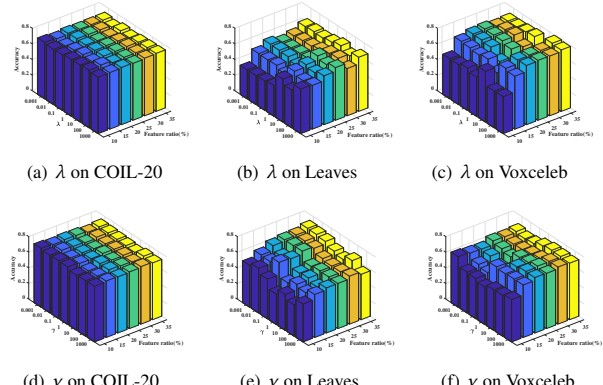

(a) $\lambda$ on COIL-20

(b) $\lambda$ on Leaves

(c) $\lambda$ on Voxceleb

(d) $\gamma$ on COIL-20

(e) $\gamma$ on Leaves

(f) $\gamma$ on Voxceleb

**Figure 5: ACC with different parameters on COIL20, Leaves and Voxceleb.**

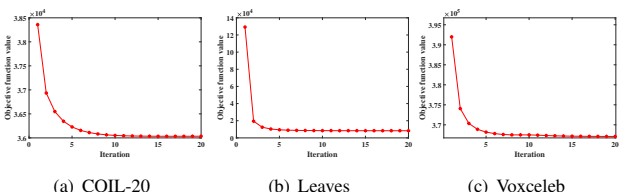

(a) COIL-20

(b) Leaves

(c) Voxceleb

**Figure 6: Variation curves of objective function values.**

## 3.5 Parameter Sensitivity and Convergence

SMUFS involves two inherent parameters, where $\lambda$ controls the smoothness of the fused membership matrix and $\gamma$ balances the influence of projecting samples to the cluster label space. To evaluate the effects of these parameters, Fig. 5 illustrates the clustering accuracy with different parameter settings and the numbers of selected features. We observe that when $\gamma$ is smaller than 1, SMUFS can achieve superior performance, which suggests that maintaining the locality of cluster labels effectively improves its discriminative capability. Meanwhile, Fig. 6 displays the objective function value versus the number of iterations, showing that the objective function decreases rapidly and converges within a few iterations, which validates the effectiveness of the proposed optimization strategy.

## 4 CONCLUSION

In this paper, we propose a novel scalable multi-view unsupervised feature selection method (SMUFS). To fully leverage the underlying structures of data, SMUFS not only fuses the aligned membership matrices as the external cluster information to obtain a unified membership matrix but also learns the similarity graph to preserve the locality of the learned cluster labels, such that the interaction between similarity structure and clustering structure can enhance the discriminative of the membership matrix, facilitating the selection of informative features. Further, SMUFS incorporates anchor strategy to reduce the computational complexity and extend its applicability to large multi-view datasets. Comprehensive experiments demonstrate the superiority of SMUFS over the state-of-the-art methods.

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
