# OpenReview forum: "Scalable Multi-view Unsupervised Feature Selection with Structure Learning and Fusion"
_acmmm.org/ACMMM/2024/Conference — MM2024 Poster_

### Official Review · Reviewer_JuFF · 2024-05-01

**Rating:** 6
**Confidence:** 4

**Summary:**

To address the challenges posed by high-dimensional data presented in multiple forms, the concept of multi-view unsupervised feature selection has gained importance as a crucial learning approach. Traditional methods typically focus on selecting features that maintain data’s similarity structure but overlook the discriminative elements within the cluster structure. Additionally, these methods often enforce orthogonal constraints on pseudo cluster labels, which can compromise the local features in the cluster label space. Furthermore, the process of learning the similarity or cluster structure from all samples can be excessively time-consuming. To overcome these issues, a method called Scalable Multi-view Unsupervised Feature Selection with structure learning and fusion (SMUFS) has been developed. SMUFS specifically utilizes sample-view weights to adaptively integrate membership matrices that represent cluster structures and act as pseudo cluster labels, facilitating the creation of a consistent membership matrix across different views to effectively guide feature selection. Additionally, SMUFS implements graph learning using the membership matrix to preserve the locality of cluster labels and enhance their discriminative power. An acceleration strategy is also included to adapt SMUFS for use with relatively large datasets. A convergent solution has been formulated to optimize the approach, and extensive testing has confirmed the efficiency and superiority of SMUFS in practical applications.

**Strengths:**

The Scalable Multi-view Unsupervised Feature Selection with structure learning and fusion (SMUFS) method presents a significant advancement in the field of multi-view unsupervised feature selection, particularly for high-dimensional data with multiple representations. The approach addresses several limitations of previous methodologies effectively, showcasing numerous strengths that make it a superior choice for such applications.

Strengths:

Unified Feature Selection Across Views: One of the primary strengths of SMUFS is its ability to adaptively integrate membership matrices that represent cluster structures from different views. This unified approach ensures more coherent and robust feature selection across all views, enhancing the overall performance of the clustering or feature selection task.
Improved Discriminative Capability: SMUFS not only focuses on preserving the similarity structure of the data but also places a strong emphasis on capturing the discriminative information within the cluster structure. By performing graph learning from a unified membership matrix, it manages to preserve the locality of cluster labels while enhancing their discriminative power, which is a considerable improvement over prior methods.
Scalability: The addition of an acceleration strategy enables SMUFS to be applicable to relatively large-scale data sets, making it more versatile and practical for real-world applications where data volume can be a challenge.
Efficiency and Efficacy: Extensive experimentation has demonstrated the effectiveness and superiority of SMUFS in handling multi-view unsupervised feature selection. Its ability to converge towards an optimal solution further attests to its efficiency in processing and analyzing data.

In summary, SMUFS presents a highly effective and scalable solution for multi-view unsupervised feature selection, offering improvements in feature selection coherency, discriminative capability, and practical scalability.

**Limitations:**

For the convenience of readers to read and understand, it is recommended that the authors provide the corresponding reference for this formula：(A+CBC𝑇)−1=A−1−A−1C(B−1+C𝑇A−1C)−1C𝑇A−1.

**Suitability:**

3

---

### Official Review · Reviewer_yArm · 2024-05-11

**Rating:** 5
**Confidence:** 4

**Summary:**

To tackle the high-dimensional multi-view data, this paper proposes a novel unsupervised feature selection method named SMUFS (Scalable Multi-view Unsupervised Feature Selection with Structure Learning and Fusion), which simultaneously utilizes the cluster structure and the similarity relations of data to identify discriminative features. Meanwhile, this method introduces the sample-view weights to fuse the aligned membership matrices and learns the graph from the unified membership matrix, not only balancing the contributions of multiple views but also preserving the locality of cluster labels. An acceleration strategy is devised to reduce the computational complexity of SMUFS, and experiments validate the effectiveness of SMUFS when handling multi-view unsupervised feature selection tasks.

**Strengths:**

1. Significance and Application: This paper focuses on the multi-view unsupervised feature selection, which has become an attractive topic with the emergence of vast amounts of unlabeled data. Feature selection can pick up the informative feature subset, thereby enhancing the performance of subsequent tasks like multimedia image clustering and classification.

2. Novelty: This paper presents a unique approach to multi-view unsupervised feature selection that fully explores the underlying structure information of data and adaptively fuses them. The interaction between cluster structure learning and similarity structure learning is interesting, and the bipartite graph learning is tactfully incorporated to improve the efficiency of the SMUFS.

3. Soundness and Clarity: The paper demonstrates its soundness by presenting a well-defined methodology and experimental validation, showing performance advantages over existing methods. Additionally, it is well-organized and easy to follow for readers.

**Limitations:**

1. Authors argue that "the l_{2,p}-norm feature selection methods require to calculate the scores of each feature and then rank the feature scores to select features", but the optimization procedure in Eq.(19) also involves the ranking of all features. Thus, what is the main difference between the l_{2,p}-norm and l_{2,0}-norm?

2. There are some typos in the current version that need to be addressed. For example, in line 416, the automatic parameter beta is related to the number of selected features k, however, Eq. 17 is unrelated to k. The authors should carefully proofread the paper and correct all the typos.

3. The alignment of membership matrices is a component of the SMUFS, thus the computational complexity analysis should consider it.

**Suitability:**

3

---

### Official Review · Reviewer_CS3M · 2024-05-13

**Rating:** 4
**Confidence:** 4

**Summary:**

The paper introduces a method called SMUFS (Scalable Multi-view Unsupervised Feature Selection with structure learning and fusion) for selecting informative features from high-dimensional multi-view data. Previous methods focused on preserving the similarity structure of the data but neglected the discriminative information in the cluster structure. SMUFS overcomes this limitation by jointly exploiting cluster structure and similarity relations. It adaptsively fuses membership matrices across views and performs graph learning to improve the discriminative capability of cluster labels. SMUFS also introduces an acceleration strategy to handle large-scale data efficiently. Experimental results demonstrate the effectiveness and superiority of SMUFS in feature selection.

**Strengths:**

1. The proposed method, SMUFS, introduces a novel approach for multi-view unsupervised feature selection that considers both the cluster structure and similarity relations of the data. It addresses the limitations of previous methods that focused solely on preserving similarity structure and neglecting discriminative information in the cluster structure. By jointly exploiting both aspects, SMUFS offers a new perspective for feature selection in multi-view data.
2. SMUFS introduces the concept of sample-view weights to adaptively fuse membership matrices across views, allowing for the generation of a unified membership matrix that guides feature selection. The method also performs graph learning from the membership matrix to preserve the locality of cluster labels and enhance their discriminative capability. These theoretical advancements contribute to a more comprehensive and effective feature selection process.
3. The paper presents extensive experiments to evaluate the effectiveness and superiority of SMUFS. The experiments cover various datasets and benchmark tasks, demonstrating the performance gains achieved by SMUFS compared to existing methods. The evaluation provides quantitative evidence of the method's efficacy and validates its usefulness in practical applications.
4. The proposed method, SMUFS, has potential applications in various fields where multi-view data analysis is crucial, such as multimedia retrieval, image processing, and document analysis. By effectively selecting informative features from high-dimensional multi-view data, SMUFS can improve the performance of subsequent tasks in these application areas. The practical relevance of the method enhances its value and applicability.

**Limitations:**

1. Some sections of the paper are poorly organized. For example, the third paragraph in the Introduction is excessively long, occupying nearly half a page. Is it really necessary to introduce all those details in a single paragraph? It is recommended to provide a concise summary of the relevant studies mentioned in the introduction, reserving detailed explanations for the main body of the paper. Additionally, Section 2 could be split into two parts, separately introducing the model and the training methods.
2. Some initialization settings are ignored, such as how to initialize the bipartite graph R and the membership matrix Z in Eq. 22.
3. The Conclusion section lacks a comprehensive discussion, particularly regarding the main limitations and difficulties encountered in the research. It is important to thoroughly explore the drawbacks and constraints of the proposed method and provide insightful suggestions for future research.
4. Overall, the experimental results presented in the paper demonstrate the effectiveness of the proposed method. However, the meaning of the red circles enclosing certain areas in Figures 4 (a)-(e) seems not explained anywhere in the text. It is unclear what aspect of the method's performance these circles represent. Additionally, why does Figure 4 (f) lack the similar red circles? And furthermore, it is important to note that the paper does not correctly reference the figures; "Fig. x" should be changed to "Figure x."

**Suitability:**

3

---

### Official Review · Reviewer_qg7p · 2024-05-23

**Rating:** 5
**Confidence:** 4

**Summary:**

This manuscript proposes a novel unsupervised Scalable Multi-view Feature Selection method, abbreviated as SMUFS. SMUFS uses the soft labels of FCM clustering as guidance for feature selection, which is an interesting idea. Additionally, to efficiently tackle the large-scale datasets, an acceleration strategy is devised, which holds practical significance.

**Strengths:**

1) SMUFS combines the unified membership matrix of FCM clustering across views with graph learning to guide the feature selection process. It leverages the intra-cluster local structure and inter-cluster discriminative power of clustering results to effectively learn representative features.
2) The authors introduce an acceleration strategy into the proposed SMUFS, effectively reducing computational complexity and enabling it to handle large-scale multi-view datasets.
3) Quantitative experiments, ablation studies, and running time experiments on 8 real-world multi-view datasets demonstrate the effectiveness of SMUFS.

**Limitations:**

1) In the experimental setup, the authors provide a detailed explanation of how the proposed SMUFS adjusts its hyperparameters. However, they do not clarify how the hyperparameters of the comparison baselines are tuned. The authors should provide the necessary details regarding this.
2) In Tables 3-4, the authors compare SMUFS with existing methods. However, the comparison lacks necessary statistical significance tests, such as presenting the average rank of each baseline method.
3) Some figures in the paper, such as Figure 3, lack clarity. The authors should make the necessary adjustments to improve their quality.

**Suitability:**

3

---

### Meta-Review · Area_Chair_hDoq · 2024-07-01

**Recommendation:** Accept (Poster)
**Confidence:** 5

**Metareview:**

This paper devises a novel unsupervised multi-view feature selection method, jointly leveraging the cluster structure and similarity relations to fully exploit the inner information among samples. Moreover, an acceleration strategy is included to handle large datasets. With clear organization and presentation, this paper presents a well-defined methodology and experimental validation, showcasing effectiveness and efficiency.

After rebuttal and discussion, all reviewers recognize the constributions and give postive scores.